# Single versus repeated heat stress in wheat: What are the consequences in different developmental phases?

**Krisztina Balla**[1]*, **Ildikó Karsai**[1], **Tibor Kiss**[2], **Ádám Horváth**[1], **Zita Berki**[1], **András Cseh**[1], **Péter Bónis**[1], **Tamás Árendás**[1], **Ottó Veisz**[1]

1 Agricultural Institute, Centre for Agricultural Research, ELKH, Martonvásár, Hungary, 2 Food and Wine Research Institute, Eszterházy Károly University, Eger, Hungary

* balla.krisztina@atk.hu

**Data Availability Statement:** All relevant data are within the manuscript and its Supporting Information files.

## Abstract

With a possible reference to heat priming and to characterize the extent and variation in the heat stress responses in wheat, the effects of single vs. repeated heat stresses were examined by measuring the changes in morphological and grain yield-related traits and photosynthetic parameters. To achieve these objectives, 51 winter wheat cultivars of various geographic origins were included in two independent experiments covering different phenological stages. In Experiment I, a single heat stress event was applied at stem elongation (SE) and booting (B), and the repeated heat stress was applied at both of these stages (SE+B). In Experiment II, the single heat stress was applied at stem elongation (SE) and full heading (CH), while the repeated heat stress was applied at both stages (SE+CH). While genotype was a more important factor for determining the morphological and yield-related traits, it was the treatment effect that mostly influenced the photosynthetic parameters, with the exception of the chlorophyll content. The heading stage was more sensitive to heat stress than the booting stage, which was primarily due to the larger decrease in the average seed number. The importance of biomass in contributing to grain yield intensified with the heat stress treatments. There was a large variation between the wheat cultivars not only in yielding abilities under control conditions but also in sensitivities to the various heat stresses, based on which 7 distinct groups with specific response profiles could be identified at a highly significant level. The 7 wheat groups were also characterized by their reaction patterns of different magnitudes and directions in their responses to single vs. repeated heat stresses, which depended on the phenological phases during the second cycle of heat stress. The possible association between these findings and heat priming is discussed.

## Introduction

Different environmental conditions, including extremely high temperatures, appear frequently during the growing season, causing significant yield losses in cereals. Currently, it is becoming more frequent that wheat varieties are subjected to different abiotic stresses several times

**Funding:** This research program was funded by a grant from the National Scientific Research Fund (NKFIH) K-119801 and by the GINOP-2.3.2-15-2016-00029 project. The funders had no role in study design, data collection and analysis, decision to publish, or prepartion of the manuscript.

**Competing interests:** The authors have declared that no competing interests exist.

during their developmental phases. It is important to know what processes take place in some developmental stages, as the different phenophases may exhibit different sensitivities to high-temperature stress [1–4]. The effect of heat stress on plants strongly depends on the timing and duration of the heat stress period during the growing season and on the genotype [5]. However, heat stress can also cause increased sensitivity among wheat varieties if they are exposed to high temperatures not once but several times at certain developmental stages. This can cause varying extents of damage to their development, physiology and yield.

Physiological processes are one of the most sensitive to high-temperature stress and in turn have significant effects on various yield components [6–9]. Lower $CO_2$ assimilation [10], increased sensitivity of stomatal closure to high temperature, and changes to the flow of $CO_2$ into the leaves are all problems that lead to a deterioration in the efficiency of photosynthesis. The inhibition of photosynthetic processes by heat stress is generally attributed to reduced rates of ribulose-1,5 bisphosphate regeneration, which is also influenced by the breakdown of electron transport activity [11]. The translocation of photosynthetic assimilates can be inhibited by the effect of heat stress, resulting in damage to the grain filling process [6]. These processes are further impaired by the accelerating effect of high temperatures [12]. The effect of long-term heat stress on the grain yield of wheat was studied during experiments across the whole vegetation period [13]. Developmental processes were accelerated by heat stress. The total aboveground biomass declined significantly by between 19% and 41%. Plants adapted to heat stress (30˚C, for 16 hours) conditions by producing less biomass. The water use efficiency was significantly decreased under heat stress conditions at the flowering and grain filling stages, resulting in reduced grain yield (-41% and -77%, respectively).

Studies have examined the effect of high-temperature stress, drought and their combination before anthesis on the growth, yield and physiological properties of some wheat varieties. Their results confirmed how detrimental preanthesis stress can be to plant development and grain yield. It has generally been shown that drought stress reduces grain yield more than heat stress does [14–16]. Nevertheless, Qaseem et al. [17] found that heat stress decreased grain yield by 53.05% compared to drought stress, where it decreased by 44.66%. The damage to chlorophyll molecules was higher under heat stress than under drought stress because both the membrane structure and the proteins disintegrated during heat stress.

However, food security needs to be maintained; therefore, it is necessary to develop the heat tolerance capacity of wheat. One solution to enhance the thermotolerance capacity of wheat plants is heat priming. Wang et al. [18] suggested that priming can be a promising strategy for crop production for plants that need to cope with abiotic stresses under global climate change. Priming is defined as the pre-exposure of plants to a stimulating factor, which could result in crop "stress memory", enabling plants to be better prepared to react to later stress events. In nature, plants are exposed to weather extremes not once but several times. One of the effects of this repeated stress is that the plants can be hardened by the weaker stresses and thus be prepared to overcome a stronger stress more easily. Pretreatment with sublethal high temperatures can contribute to achieving temporary thermal tolerance and can defend plants from subsequent higher temperatures [19–22]. Most studies, however, have focused on crop responses to single heat stress periods. Therefore, there are relatively little and sometimes divisive data on the effect of double heat stress (heat priming) in wheat.

Fan et al. [23] evaluated the effect of heat priming applied during the stem elongation, booting and anthesis stages, followed by 5 days of heat stress during the grain filling stage. Their results confirmed that heat priming at early reproductive stages can improve postanthesis heat tolerance and that heat priming at the booting stage was more effective than heat priming at the stem elongation stage or anthesis [23]. When comparing the nonprimed plants to the heat-primed plants, the data showed that heat priming (at the stem elongation stage and booting)

prevented the grain yield injuries caused by heat stress during grain filling. The photosynthetic activity, stomatal conductance and chlorophyll content increased in the heat-primed plants in comparison with nonprimed wheat cultivars. Among the heat priming treatments, heat priming at booting + heat stress at the grain filling stage resulted in the greatest net assimilation level at the late grain filling stage compared to the control. Tolerance may have improved due to the reduced level of reactive oxygen species (ROS) and increased activity in antioxidant enzymes, such as superoxide dismutase and peroxidase.

Priming also has an effect on photosynthetic processes [24, 25]. Decreased flag leaf photosynthesis and antioxidative enzymes and enhanced cell membrane peroxidation and $O_2^{\bullet-}$ were found under both pre- and postanthesis heat stress, but under postanthesis heat stress, plants with preanthesis heat priming (acclimation) showed much higher photosynthetic activity than those without preanthesis heat stress [24]. Not only photosynthetic processes but also the chlorophyll a/b ratio and antioxidant enzyme activities were positively affected by preanthesis heat stress. Multiple heat priming events may have contributed to increased thermal tolerance to later high-temperature stress, as exemplified by the higher activity of antioxidant enzymes (e.g., superoxide dismutase and glutathione reductase) in the chloroplast maintaining better redox homeostasis in the plants that received heat priming [25]. The upregulated gene expression of Rab, Cu/Zn-SOD, Mn-SOD and GR contributed to the improvement of photosynthetic processes and antioxidant capacity in primed plants [24, 25]. Wang et al. [26] showed that high-temperature acclimation could relieve the damage to stem-stored carbohydrates and grain starch remobilization due to heat stress at anthesis. Carbohydrate remobilization showed an increase due to preanthesis heat stress acclimation. In addition, there was an improvement in the grain yield of pretreated plants compared to wheat plants that were exposed to postanthesis heat stress only [26].

Wollenweber et al. [27] described that they did not find an association between the plants exposed to heat stress at vegetative and reproductive developmental stages. Their wheat variety was investigated at the double-ridge stage (DR), anthesis (AN) and a combined DR+AN stage. The applied heat stress at the double-ridge stage did not affect the grain dry weight and grain number at the DR+AN developmental stage when exposed to heat stress. The grain number decreased by 41% and the grain dry weight decreased by 40% due to heat stress applied at the DR+AN stages, similar to the results for heat stress at anthesis alone. Mendanha et al. [28] also did not obtain convincing results about the priming effect of heat stress at anthesis. One of the varieties studied did not show differences in the assimilation rate and yield parameters between the primed and nonprimed treatments.

In this research project, our aim was to define how repeated heat stresses influence yield-related traits and physiological parameters compared to single heat stress treatments in winter wheat varieties and what connection can be found with heat priming. Single and repeated heat stress was studied at different developmental phases in two independent experiments under controlled conditions. The uniqueness of our project was that a larger number of wheat varieties were tested, in contrast to other experiments where only a few varieties were examined [13, 23–26, 28, 29]. In both heat stress experiments, 51 winter wheat varieties of different geographic origins were investigated by measuring various photosynthetic parameters and yield-related traits under controlled environmental conditions in the greenhouse and phytotron facilities of Martonvásár.

## Materials and methods

### Crop management

A total of 51 winter wheat varieties with different geographic origins (S1 Table) were included in a series of experiments under controlled conditions in a greenhouse and a phytotron to

study their responses under single and repeated heat stress conditions applied at different developmental stages. The heat stress responses of the wheat varieties were determined in two independent experiments, in which the same standard plant raising protocols were applied. The changes in the physiological processes and the agronomic traits were examined based on heat stress treatments applied separately (in one phenophase) or in combination (in two phenophases) across various developmental stages (Fig 1).

## Experimental conditions and design

Two experiments were conducted at the Agricultural Institute, Centre for Agricultural Research, Hungary. Winter wheat varieties (*Triticum aestivum* L.) were grown in pots filled with approximately 1.5 kg of a 3:2:1 mixture of garden soil, compost and sand. The germinated seedlings were vernalized in peat blocks for 60 days at 4˚C with a low light intensity and short day length, and then the plants were transferred to individual pots. The plants received daily watering and a twice-weekly supply of nutrients (Volldünger Solution, Linz, Austria, in tap water).

After the vernalization treatment, the wheat plants were raised in a greenhouse under relatively standard conditions, where the ambient temperature ranged between 25˚C (day) and 19˚C (night) and the natural light conditions were supplemented with artificial light at 170 µmol $m^{-2}\,s^{-1}$ intensity produced by metal halide lamps to reach a 16-hour photoperiod

## Developmental stages

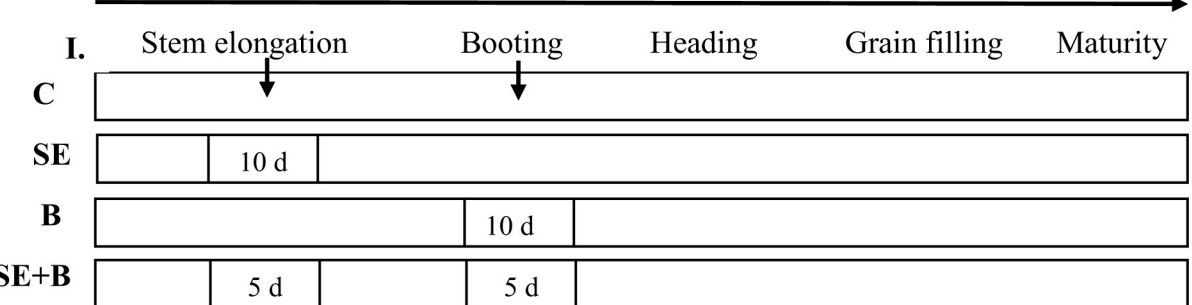

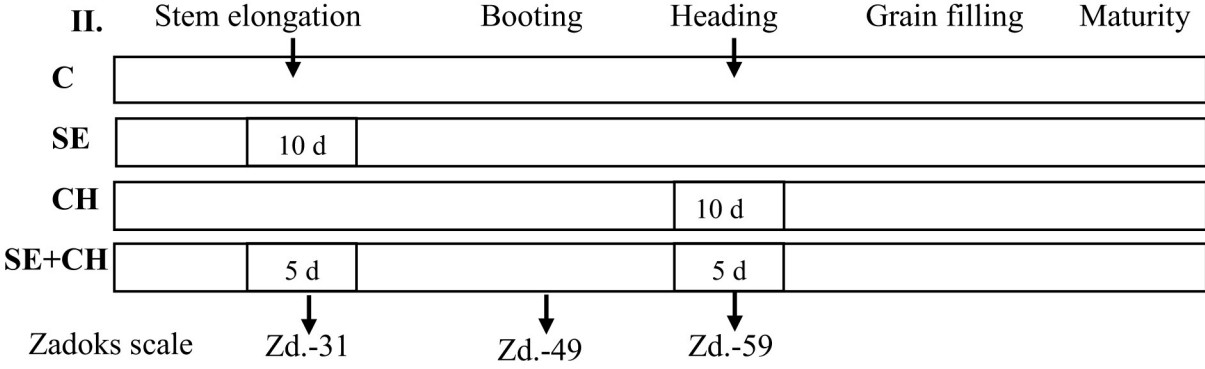

**Fig 1. The experimental design of the two heat stress experiments.** C refers to no heat stress treatment-control; SE refers to 10 days of heat stress treatment at the stem elongation stage; B refers to 10 days of heat stress treatment at the booting stage; CH refers to 10 days of heat stress treatment at full heading; SE+B refers to 5 days of heat priming at the stem elongation stage + 5 days of heat stress at the booting stage; and SE+CH refers to 5 days of heat priming at the stem elongation stage + 5 days of heat stress at full heading.

regime per 24-hour cycle. In both individual experiments, 16 plants of each genotype were raised in individual pots in the greenhouse and rotated regularly during monitoring of their developmental patterns. Twelve of the original 16 plants with the most similar developmental and phenological aspects were then selected and included in the stress experiment, with 3 plants per treatment as replications.

Control plants of each variety were raised in the greenhouse throughout their life cycle, while the pre-stressed plants of each variety at the given phenophase were transferred to the heat-stress chamber (Conviron PGV-36, PGR-15 growth chambers, Conviron Ltd., Canada) in the Martonvásár phytotron for a given stress period (10 days or 5+5 days). At the end of the treatments, the wheat plants were returned to the greenhouse and raised with the control plants until maturity.

In the **first experiment** (**Exp. I.**), one control and three heat stress treatments were applied at the different developmental stages: 1. Control, 2. Stem elongation (SE, the first node was detectable), 3. Booting stage (B), and 4. Stem elongation + booting stage (SE+B) (Fig 1).

In the **second experiment** (**Exp. II.**), one control and three heat stress treatments were applied at the following phenophases: 1. Control, 2. Stem elongation (SE), 3. Full heading (CH), and 4. Stem elongation + full heading (SE+CH).

In the heat stress chambers, the plants were kept under a 16-hour photoperiod regime, and a light intensity of 350 µmol m$^{-2}$ s$^{-1}$ was produced by metal halide lamps. The relative humidity (RH%) was set to 64–68% during the day and 76% at night in the stress chamber. The applied temperature values of the heat stress treatments were the same in both experiments (Table 1). Day/night temperatures of 28/20°C were applied at the early developmental phase (SE), and 36/20°C was set at the later stages (B, CH). The day temperature gradually increased from the night temperature (20°C) to 28 or 36°C, which was maintained for 8 hours and then gradually decreased to 20°C.

The heat stress treatments lasted for 10 days for the single treatments in one phenophase, such as the stem elongation, booting and full heading stages, but treatments lasted 5+5 days for the double stress treatment across two phenophases. The phenophases of the plants were monitored every day and determined based on the double-digit system of the Zadoks scale [30]. As a result of monitoring development, every wheat plant received heat stress treatment in the same specific developmental stage examined within the given experiment.

**Table 1. Temperature values for the photosynthetic measurements in LI-6400 and temperature values of the stress chambers.**

| Experiment I. | | | C | SE | B | SE+B |
|---|---|---|---|---|---|---|
| | | | | (Zadoks-31) | (Zadoks-49) | (Zd. 31+49) |
| Temperature inside the LI-6400 leaf chamber | Control conditions | | 19–25°C | 19–20°C | 22–25°C | 22–25°C |
| | Heat stress conditions | | - | 28°C | 36°C | 28+36°C |
| Temperature of the growth chambers in the phytotron | Heat stress conditions | | - | 28°C | 36°C | 28+36°C |
| Number of days of treatment | | | | 10 days | 10 days | 5+5* days |
| **Experiment II.** | | | **C** | **SE** | **CH** | **SE+CH** |
| | | | | (Zadoks-31) | (Zadoks-59) | (Zd. 31+59) |
| Temperature inside the LI-6400 leaf chamber | Control conditions | | 19–25°C | 19–20°C | 22–25°C | 22–25°C |
| | Heat stress conditions | | - | 28°C | 36°C | 28+36°C |
| Temperature of the growth chambers in the phytotron | Heat stress conditions | | - | 28°C | 36°C | 28+36°C |
| Number of days of treatment | | | | 10 days | 10 days | 5+5* days |

C: control, SE: stem elongation, CH: full heading, SE+CH: stem elongation+ full heading; *during the double stress treatment, the photosynthetic activity was measured only on a single occasion after the second 5-day heat stress treatment, when the plants received the heat stress treatment at the booting/heading stage.

## Morphological measurements

A total of 31 traits were examined in both experiments (S2 and S3 Tables), from which 8 traits were selected for more detailed analysis. The morphological parameters and yield-related traits were measured after the plants reached harvest maturity. The morphological properties included measurements of plant height (PH), the length of the last internode (LIN), total aboveground biomass (straw + all ears, FBIOM) and grain yield per plant (GY). The harvest index (HI), grain number per spikelet (SPS), thousand-kernel weight (TKW) and average seed number (AS) were calculated from the received data.

## Physiological measurements

Among the physiological parameters, the chlorophyll content (CLR) was measured on a single occasion after the heat stress treatment using a SPAD-502 instrument (Minolta, Japan), which records leaf transmittance in the red and near-infrared spectra and then calculates the SPAD index from these two values. Three wheat plants as replicates (per treatment) were measured for chlorophyll content and for gas exchange measurements. The last fully expanded leaves were used for measurements during heat stress at the stem elongation stage and booting. However, at anthesis, the analysis was performed on the middle of the flag leaf.

Changes in the net assimilation rate (PN), transpiration (E), stomatal conductance (GS) and intercellular $CO_2$ concentration (CI) of the plants were measured using an LI-6400 Portable Photosynthesis System (LI-COR, Inc., Lincoln, NE, USA). The infrared gas analysis system was equipped with a leaf cuvette that exposed 6 $cm^2$ of the leaf area. The leaves were kept in a leaf chamber during the measurements. External air was scrubbed of $CO_2$ and mixed with a supply of pure $CO_2$ to create a reference concentration of 390 µmol $m^{-2}$ $s^{-1}$. The $CO_2$ concentration was maintained at a constant level using a $CO_2$ injector with a high-pressure $CO_2$ gas cartridge source. The quantum flux was set to 300 µmol $m^{-2}$ $s^{-1}$, and the flow rate was set to 300 µmol $m^{-2}$ $s^{-1}$. The temperature inside the leaf chamber was maintained at 18–20˚C under control conditions at the stem elongation stage in both experiments and 20–22˚C at the booting and full heading stages in the first and second experiments, respectively. The temperature inside the cuvette was 28˚C at the stem elongation stage and 36˚C at the booting and heading stages under heat stress conditions in both experiments. The photosynthetic parameters were determined on the final day of the heat stress treatment, which lasted for 10 days under single heat stress conditions. During the repeated heat stress treatment, the photosynthetic properties were measured on a single occasion after the second 5-day heat stress treatment, when the plants received the heat stress treatment at the booting/heading stage.

## Statistical analysis

The data were statistically evaluated using two-way analysis of variance (ANOVA) with replications using the GenStat® (VSN International Ltd. 18th ed.) software packages. To identify the effects of the heat stress treatments and genotypes in explaining the phenotypic variance of the traits measured, a mixed linear model (REML) was used. In estimating variance components ($\sigma^2$), all effects (genotype (G) and treatment (T)) were considered random to be able to estimate the factor interactions. The correlation among traits was calculated using the mean value for each genotype and 'corrplot' package in R statistics (v.3.6.3).

To distinguish the possible patterns in the genotypic heat stress responses, the K-means clustering protocol of the STATISTICA software package, version 13.5.0.17 (TIBCO Software Inc), was applied to the data matrix of the 51 wheat cultivars, including the following 6 phenotypic traits: GY-Cav (averages of the grain yield of the two controls in Exp. I. and II.), GY_SEav (averages of the grain yield of the two SE treatments in Exp. I. and II.), GY_B and GY_SE+B

from Exp. I., and GY_CH and GY_SE+CH from Exp. II. The option of maximizing the initial between-cluster distances was set, and the most likely cluster number was evaluated in the range of 2 to 10 clusters, with 10 iterations for each round. The most likely cluster number was determined based on the changes in the sum of the within-cluster distances from the cluster means at each cluster number increase. The correctness of the established cluster number was checked on the same data set by applying the forward stepwise module of discriminant analysis in STATISTICA software using the cluster positions of the cultivars as a dependent variable and the six GY parameters as independent variables.

## Results

### Overall effects of single and repeated heat stress on yield-related traits of the wheat genotypes

In the experimental setup of 51 wheat genotypes × heat stress treatments at different developmental phases, all the examined traits were significantly influenced by the two factors but to different extents across the two experiments (S2 and S3 Tables). In general, the morphological and yield-related traits were mostly determined by the genotype in Exp. I., while in addition to the genotypic effect, the treatment effect was more pronounced in Exp. II. In both experiments, the strongest genotypic effect was observed for morphological traits, which explained between 53.74% (LIN) and 76.07% (PH) of the phenotypic variance in Exp. I. and between 63.26% (LIN) and 82.57% (PH) in Exp. II. In Exp. I., the treatment effect on yield-related traits was significant, but its role was smaller, contributing between 1.94% (SSPIK) and 26.77% (SPS) to the phenotypic variance, while the genotypic effect was more pronounced, explaining 29.68% (SPS) and 82.52% (BIOM) of the phenotypic variance. In Exp. II., the heat stress treatment significantly influenced GN, GNSP, GY, AS, ASW, SEAW, SSN, and SSW, explaining between 25.9% (SEAW) and 43% (AS) of the phenotypic variance. The heat stress treatment was the most significant component in determining both the seed number and seed weight and was more prevalent in side ears. In the case of physiological parameters, however, the effect of treatment became the most decisive factor in both experiments; the only exception was CLR, where the effect of genotype was strongest in Exp. I., but the effect of treatment was strongest in Exp. II.

The control treatments of the two independent experiments were used as technical replications (Figs 2 and S1). There were no significant differences between the control treatments of the two experiments in plant height, last internode length, total aboveground biomass, grain yield, thousand kernel weight or grain number per spikelet. Most of the other traits examined also showed no significant differences (S1 Fig). The similarity of the results emphasizes that the plants grew equally in the two experiments, making it possible to compare and merge the two data sets.

Averaged over the 51 wheat varieties, the overall response to heat stress showed significant decreases in the eight selected morphological and yield components at the various developmental stages (Fig 3).

The extent of the reduction, however, depended on both the phenophase and the number of heat stress cycles. In both experiments, it was shown that the single heat stress applied at the stem elongation stage (SE) caused the least significant or nonsignificant changes in the studied properties. The application of a single high-temperature treatment in the booting (B) or heading stage (CH) more adversely affected the plants, which could be partly due to the differences in the temperature levels applied (28˚C vs. 36˚C). The average values from the repeated heat stress treatments were generally closer to those of the single heat stress treatments at the two later developmental phases, but significant improvements could be detected in several traits.

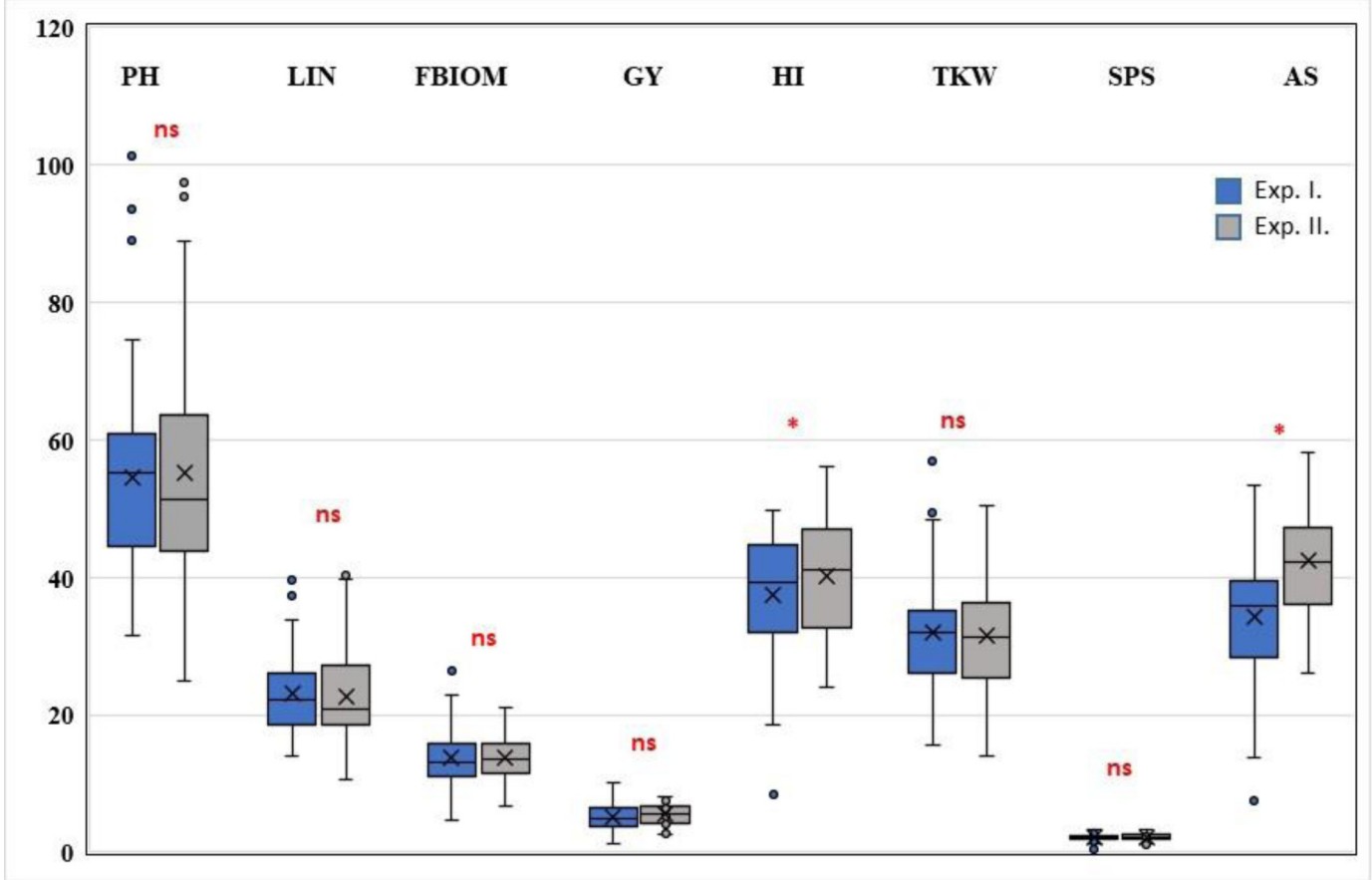

**Fig 2. Comparison of various examined properties in the control treatments of the two heat stress experiments.** PH—Plant height, LIN—Last internode length, FBIOM—Total aboveground biomass (straw + all ears), GY—Grain yield, HI—Harvest index, TKW—Thousand kernel weight, SPS—Grain number per spikelet, and AS—Average seed number; ns—not significant and *—significant at the P≤ 0.05 level.

In Exp. I. (Fig 3), the single heat stress at the stem elongation stage did not cause significant changes in the last internode length (LIN), total biomass (FBIOM), the harvest index (HI), thousand kernel weight (TKW) or grain yield (GY). During the booting stage, however, significant decreases were observed for most of the traits, the extent of which was the smallest for plant height (13.9%) and the harvest index (14.6%) and the largest for seed number per spikelet (36.07%) compared to the control. The large reduction in average seed number (34.2%) was similarly strong as that for SPS, but this reduction was partially compensated by increased TKW (+13.9%), resulting in a final 27.3% decline in grain yield compared to the control. The reduction in average seed number (AS) was accompanied by the smallest reductions in the harvest index and biomass (15%). Among the morphological features, LIN was more strongly affected by heat stress treatment, with a 27.4% reduction. By comparing the results of double heat stress (SE+B) to the single stress at booting (B), we observed significant increases in PH, LIN, AS, FBIOM, HI and GY.

In Exp. II. (Fig 3), high temperature at the stem elongation stage had somewhat stronger effects on some of the traits than those in Exp. I. These effects included decreases in GY, TKW, HI, and FBIOM and increases in PH and LIN. In addition, wheat plants suffered more from heat stress during heading than at the booting stage, as evidenced by the larger decreases in GY (43%), HI (29.4%) and AS (55%) compared to the control. Heat stress treatment applied at

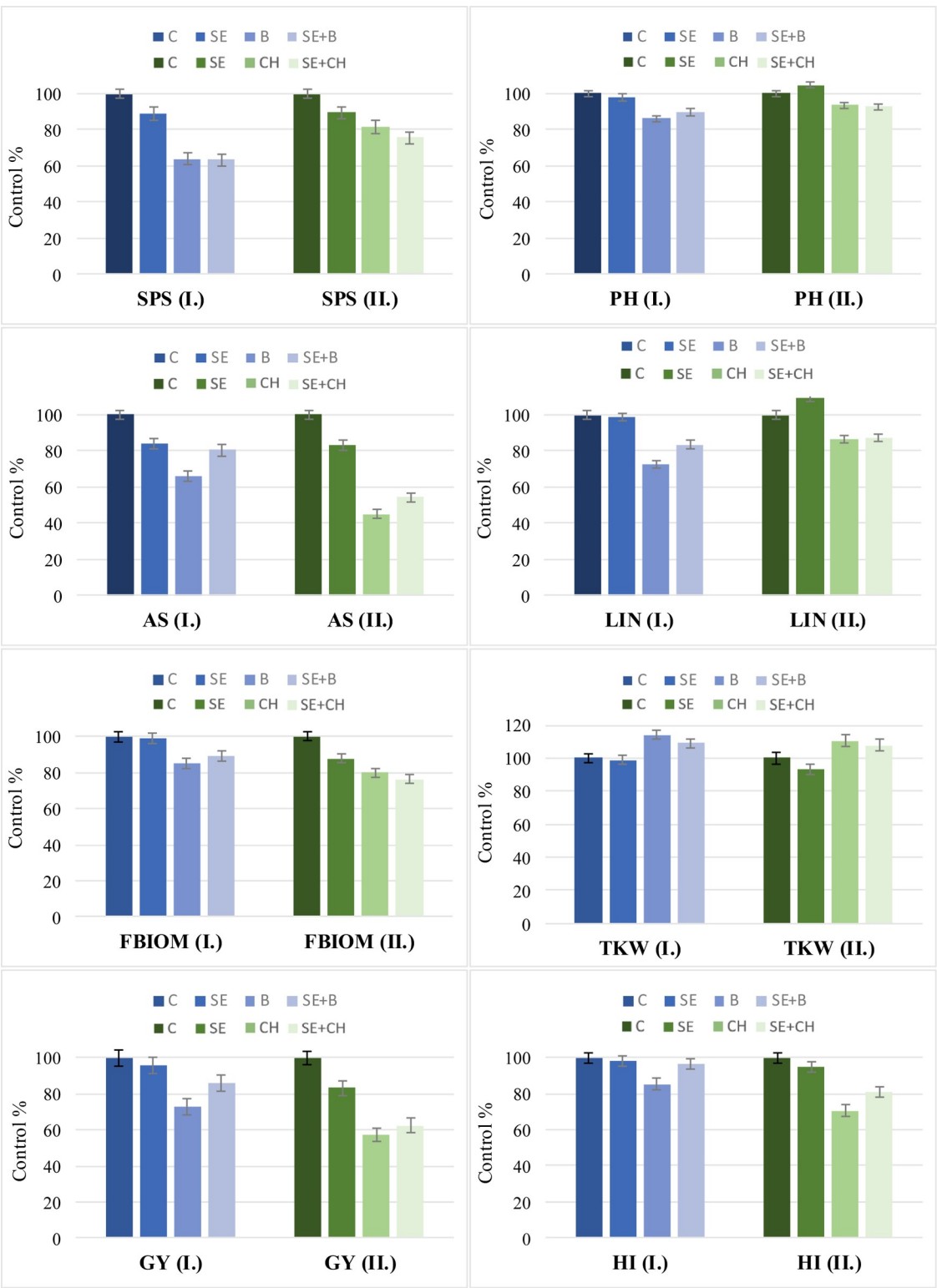

**Fig 3. Effects of single and repeated heat stress treatments on various yield-related traits in two experiments.** The results are averaged over 51 wheat cultivars at different developmental stages. I.: Experiment I., II.: Experiment II.; C: Control, SE: Stem elongation, B: Booting, CH: Full heading, SE+B: Stem elongation + booting; SE+CH: Stem elongation + full heading, GY: Grain yield per plant, AS: Average seed number, HI: Harvest index, TKW: Thousand kernel weight, FBIOM: Total aboveground biomass (straw + all ears), PH: Plant height, LIN: Last internode length, and SPS: Grain number per spikelet. The data are means ± LSD (n = 51). Least significant differences (LSD$_{5\%}$) denoted by bars on the columns represent significance at the P≤ 0.05 level.

the heading stage caused smaller changes in PH (6.4%), LIN (13.6%) and SPS (18.3%) than that applied at booting. Although the strong decrease in AS was again partially compensated by increased TKW, as in Exp. I., but in this case, it was not enough to prevent a significant decrease in GY.

The possible priming effect of heat applied at the stem elongation stage prevailed in both experiments and was clearly visible in the case of the double heat stress treatments (SE+B and SE+CH). If we compare B with SE+B (Exp. I.) and CH with SE+CH (Exp. II.), the differences achieved due to the heat priming can be clearly seen: Exp. I.: GY: +17.8%, AS: +22%, and HI: +13.3%; Exp. II.: GY: +9.5%, AS: +20.5%, and HI: +14.3%.

In the control treatments of both experiments, there were strong, significantly positive correlations between the grain yield and all the other morphological and yield-related traits (S2 and S3 Figs). When applying the various heat stresses, in general, the effects of AS, HI and FBIOM increasingly determined the GY, especially in the most severe stress treatments, while that of the TKW decreased to various extents.

## Overall effects of single and repeated heat stress on photosynthesis-related traits of wheat genotypes

In the case of photosynthetic properties, the heat treatment factor was more decisive than the genotypic effect in both experiments (S2 and S3 Tables), explaining large portions of the phenotypic variance, especially for net assimilation (PN: 51.71% in Exp. I. and 34.95% in Exp. II.) and evaporation (EVP: 41.55% in Exp. I. and 55.41% in Exp. II.). The chlorophyll content (CLR) was the only exception, for which the genotype factor was more important (24.43% in Exp. I. and 35.11% in Exp. II.).

The overall responses of the wheat varieties based on the photosynthetic properties were similar across the two experiments (Fig 4).

The data are means ± LSD (n = 51). Least significant differences (LSD$_{5\%}$) denoted by bars on the columns represent significance at the P$\leq$ 0.05 level.

The chlorophyll content was influenced to the smallest extent by heat stress. In Exp. I., smaller increases were detected during both heat stress treatments, while in Exp. II., smaller decreases were shown at the heading (CH) and SE+CH stages compared to the control. On the other hand, PN, GS and CI showed the strongest decrease due to the heat stress treatments, while E increased to a great extent. For E, the responses to heat stress increasingly intensified in the single heat stress treatments from the early developmental stage (SE) to the later phenophases (B or CH). Heat stress had a greater effect on transpiration at the booting stage (E: +-98.13%) than at heading (E: +69.9%) compared to the control. Double heat stress elicited even more enhanced transpiration responses in both experiments. In the case of GS, the heat stress at the stem elongation stage caused a smaller reduction that further dropped at the booting (GS: 58.06%, the extent of changes compared to the control) and heading stages (51.83%). For PN, the responses of plants to heat stress were the strongest at the booting and heading stages, showing a 31.6% reduction at the booting stage and a 19.68% reduction at the heading stage compared to the control.

A possible priming effect due to heat stress applied at the stem elongation stage could be detected for PN and E. In the case of PN, the extent of the increase in response to repeated heat stress was +13.6% at SE+B and +4.3% at SE+CH compared to the B and CH stages, respectively, while for E, these values were +16.2% at SE+B and +2.6% at SE+CH (Fig 4).

In general, the photosynthetic traits showed positive relationships with each other, which intensified under heat stress conditions, while the chlorophyll content was independent of the other traits in most of the cases (S2 and S3 Figs). A significant positive correlation existed

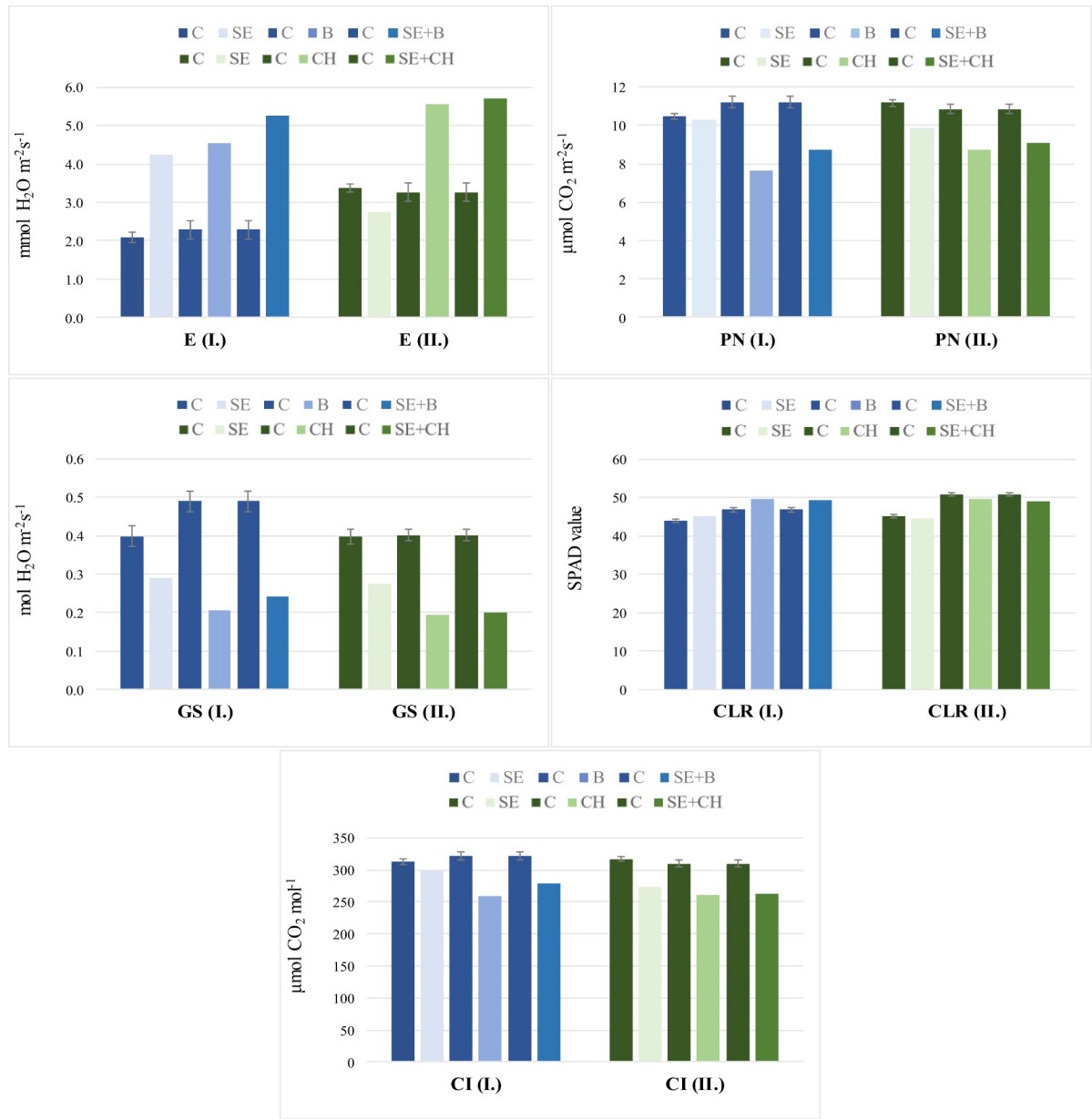

**Fig 4. Effects of single and repeated heat stress treatments on photosynthetic parameters in two independent experiments.** The results are averaged over 51 wheat cultivars at different developmental stages. I.: Experiment I., II.: Experiment II.; C: Control, B: Booting, SE+B: Stem elongation and booting, CH: Full heading, SE+CH: Stem elongation and full heading, PN: Net assimilation, GS: Stomatal conductance, E: Transpiration, CI: Intercellular $CO_2$ concentration, and CLR: Chlorophyll content.

between GS and CI, GS and E, and E and CI in both experiments. The associations between the photosynthetic parameters and the various yield-related traits, on the other hand, were very sparse, and in addition, they could be characterized by changeable patterns between the two experiments.

In Exp. I., the correlations between some of the photosynthetic parameters and yield components were evident mostly in the control treatment; these correlations diminished in the various heat stress treatments (S2 Fig). FBIOM of the yield components showed the most

significant correlation with PN (positive), GS (negative), CI (negative) and E (negative) under control conditions. Of these, the correlation between GS and CI remained significant only under the two single heat stress treatments. FBIOM negatively correlated with GS (r = -0.286* at the SE stage and r = -0.329* at the B stage) and with CI (r = -0.418** at the SE stage, r = -0.51*** at the B stage). In the case of repeated stress, however, none of these associations were significant. CLR, independent of the photosynthetic parameters, showed a significant correlation with HI (r = 0.521***) in the control treatment. At the SE+B stage, the correlations between the yield-related and photosynthetic traits generally diminished, but most of the significant relationships of CLR with yield-related traits became apparent in this treatment; CLR correlated with FBIOM (r = -0.289*), HI (r = 0.305*) and TKW (r = 0.332*).

In Exp. II. (S3 Fig), the correlation analysis again revealed relatively strong associations among the various photosynthetic traits, while a very low level of association existed between yield components and photosynthetic traits under control conditions. In contrast, the most significant (negative) correlations were shown under heat stress conditions at the stem elongation stage (SE) and heading stage (CH). PN had a strong correlation with GS (r = 0.596*** at CH, r = 0.592*** at SE+CH), CI (r = 0.49*** at CH and r = 0.424** at SE+CH) and E (r = 0.613*** at CH and r = 0.67*** at SE+CH). The strongest correlation was found between E and GS (r = 0.966*** at CH and r = 0.9*** at SE+CH). Some correlations appeared between yield-related and photosynthetic traits, which became more apparent under repeated heat stress. CLR showed closer significant correlations with HI (r = 0.368**) and SPS (r = 0.394**). PN had a significant negative correlation with PH (r = -0.311*) and FBIOM (r = -0304*).

## Genotypic response profiles to single and repeated heat stresses at different developmental phases

In the variance analyses, the effect of genotype was a significant component in determining most of the yield-related traits in both experiments (S2 and S3 Tables). This was also the case for grain yield, where 45.4% and 26.5% of the phenotypic variance was explained by the genotype in Exp. I. and II, respectively. To determine whether it is possible to distinguish specific heat stress response profiles in this group of wheat cultivars, further exploratory analyses were carried out on the grain yield data matrix of the 51 cultivars, which combined the information from both experiments. With the K-means clustering protocol, the presence of 7 distinct groups could be established as the most likely cluster number (S4 Table), which was also confirmed by the discriminant analysis, showing a 100% overlap between the observed and predicted classifications at a highly significant level (p<0.0000). The numbers of genotypes belonging to each cluster were in the range of 5 to 12, and the majority of these 7 clusters formed separate groups in the scatterplot, especially Groups 2, 4, 6 and 7 (Fig 5). There was only a slight overlap between Groups 1 and 3.

The grain yield characteristics of the 7 groups across the various heat stress treatments are illustrated in Fig 6.

The single high temperature treatment at the stem elongation phase (SE) caused small and or nonsignificant changes in GY in most groups compared to the control (Fig 6A). On the other hand, the groups showed variance in their heat stress response depending on both the developmental stage and the number of heat cycles. The four distinctly separated groups were characterized by the most specific reaction profiles. Group 2 had the overall lowest grain yield across all the treatments, and it was among the more sensitive clusters for most of the heat stress treatments, although there were no significant differences between the GY reached under the various stress conditions. Groups 5, 6 and 7 gave the highest GY under control conditions, but they showed differences in their response profiles to heat stresses applied at various

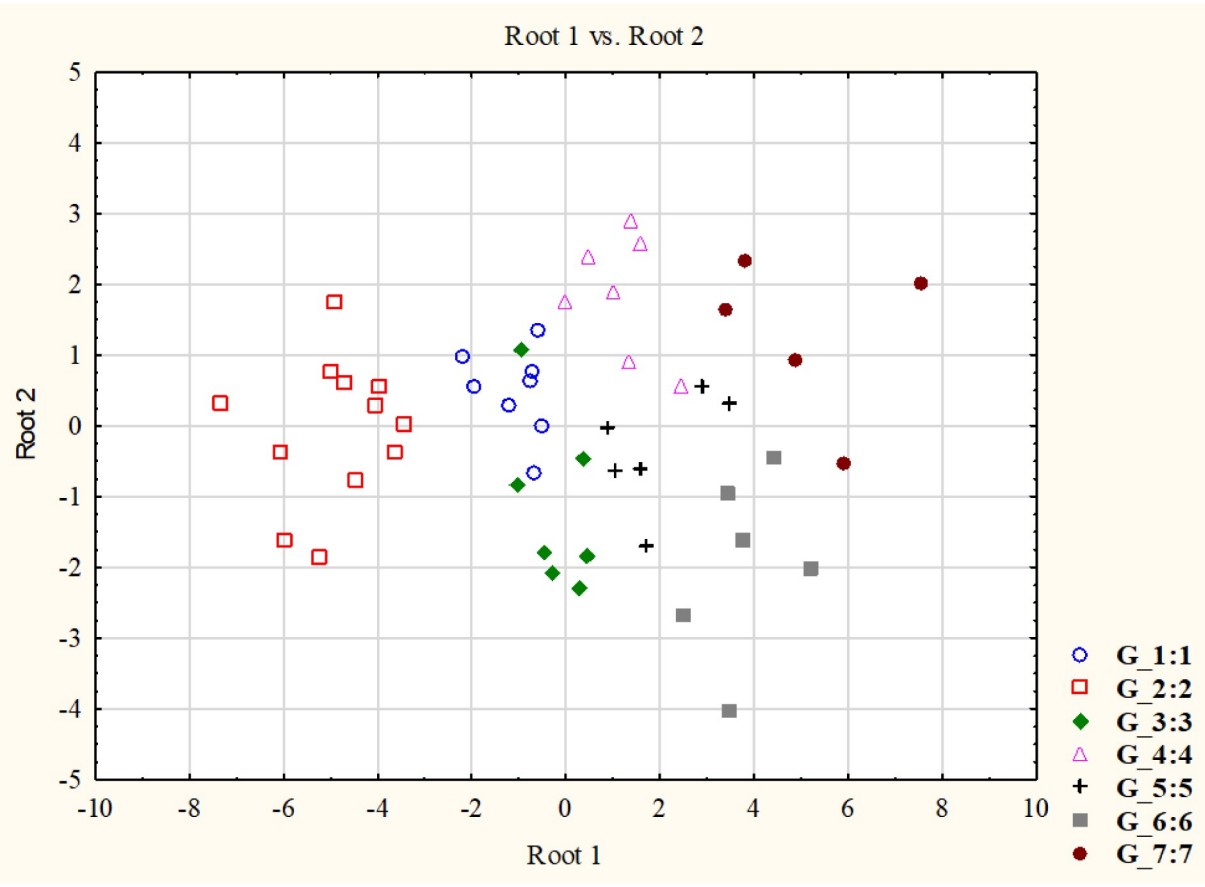

**Fig 5. Discriminant analysis of the 7 phenotypic clusters of 51 wheat genotypes.** The seven phenotypic clusters were established by K-means clustering of the grain yield data matrix, which was derived from the various heat stress treatment of Experiments I. and II.

developmental stages. The wheat cultivars in Group 7 were the least sensitive to heat stresses at the booting stage (B), producing grain yields similar to the control. Groups 5 and 6, on the other hand, were very sensitive to heat stresses, especially Group 5, which, with the exception of the repeated heat stress at booting, was one of the most sensitive groups when compared to the control across all the other stress treatments. Of the three groups (Groups 1, 3, and 4) with intermediate GY under control conditions, Group 3 showed a good level of heat stress tolerance when compared to the control, irrespective of the developmental phase. A similar case was found for Group 4, with the exception of the single heat stress treatment at full heading. The response of Group 1 was specific based on the significantly higher GY during repeated heat stress treatments irrespective of the developmental phases.

The groups showed large differences in their responses to repeated heat stress (Fig 6B). Five-day exposure to high temperature at the beginning of stem elongation significantly influenced heat stress tolerance in the later developmental phases in most cases, the magnitude and direction of which, however, proved to be dependent on both the wheat groups and the developmental phases. Only in Group 1 was the heat stress tolerance significantly enhanced in both later developmental phases as a result of the high temperature pretreatment. Groups 2, 5, and 6 responded positively to the high temperature pretreatment only in the case of heat stress applied at the booting stage, but this stimulating effect was much smaller (Groups 5 and 6) or not significant (Group 2) at full heading. In contrast, for Groups 4 and 7, the stimulating effect

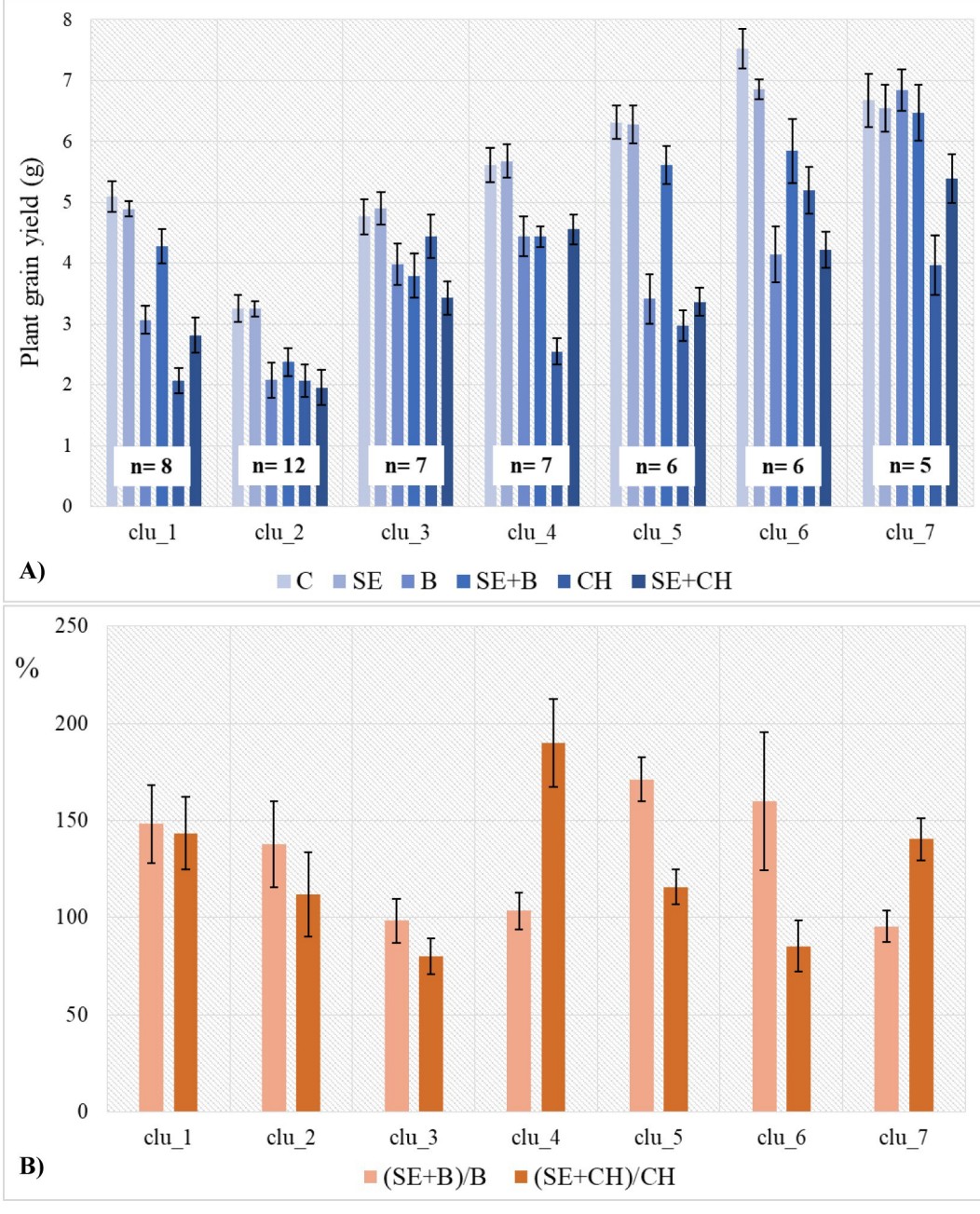

**Fig 6. Grain yield profiles of the 7 phenotypic clusters of 51 wheat genotypes.** (A) across the various heat stress treatments of Experiments I. and II. and (B) measured by the effect of repeated heat stress to the single heat stress treatment in two developmental phases expressed as % of change. C (averages of the grain yield of the two controls in Exp. I. and II.); SE (averages of the grain yield of the two SE treatments in Exp. I. and II.); B (grain yield of the single heat stress at the booting stage) and SE+B (grain yield of the repeated heat stress at the stem elongation and booting stages) from Exp. I.; and CH (grain yield of the single heat stress at full heading) and SE+CH (grain yield of the repeated heat stress at stem elongation and full heading) from Exp. II.

of the high-temperature pretreatment was only significant when it occurred before full heading but not at booting. The wheat genotypes in Group 4 showed the strongest enhanced heat stress tolerance among all the groups; plants pretreated with high temperatures at stem elongation showed a 90% increase in grain yield at full heading compared to those under single heat stress.

Group 3 was the only cluster that did not respond positively to the pretreatment in either stage; additionally, at booting, the pretreatment had no significant effect, but at full heading, its effect became detrimental, causing a significant decline in the heat stress tolerance level. The most characteristic members of each group are listed in Table 2. Neither of the groups showed separation based on the geographic origins of the wheat cultivars.

## Discussion

Our study is part of a long-term research programme in which we have already investigated and defined how sensitivity to heat stress changes with plant development and stress duration [5]. Based on our previous results, 51 genotypes were selected to further examine the effects of repeated heat stresses at the most sensitive phenophases. In two independent experiments, the selected wheat varieties were subjected to high temperature stress not only in one developmental stage separately but also in combinations of two successive phenophases. The experiments were performed under controlled conditions, and each genotype was exposed to heat stress at exactly the same developmental stage. In both experiments, the plants developed approximately uniformly, which was underlined by the comparable results of the control treatments, due to which the data could be combined in the analyses.

The timing of the first early heat stress was chosen to be at the initiation of stem elongation for the following reasons. It is a critical developmental phase in terms of yield for leaves, roots and inflorescences, as growth is very rapid, and it is characterized by a large increase in the volume of vegetative organs [31]. The existing environmental conditions determine the process of organ formation and thus the number and size of the reproductive organs [32, 33]. Kiss et al. [34] found in their development dynamics research that temperature did not affect plant development proportionally; it influenced stem elongation to the largest extent, and warmer temperatures lengthened the lag phase between the detection of the first node and the beginning of intensive stem elongation. In addition, we wanted to ensure a lag period between the two stresses for recovery, which may become increasingly difficult if the occurrence of the two developmental phases is closer in time, especially as the time elapsing between successive stages is shortened by higher temperature [33, 35]. The same aspects were taken into consideration when choosing the temperature level for the first cycle of heat stress treatments. The 28°C used in our experimental setup at this phase does not represent real heat stress, but it already lays in the supra-optimal range, well above the optimal temperature of wheat [2, 35]. The choice of 28°C was made as a compromise between not deviating much from natural weather conditions and being high enough to initiate a stress response. It has been shown that

**Table 2. The most characteristic members of the 7 phenotypic wheat groups identified via K-means clustering and discriminant analyses based on their grain yield profiles.**

| Group 1 (n = 8) | Group 2 (n = 12) | Group 3 (n = 7) | Group 4 (n = 7) | Group 5 (n = 6) | Group 6 (n = 6) | Group 7 (n = 5) |
|---|---|---|---|---|---|---|
| ave dist = 0.601 | ave dist = 0.785 | ave dist = 0.768 | ave dist = 0.597 | ave dist = 0.666 | ave dist = 0.809 | ave dist = 0.830 |
| Divana (HR) | Feng-you-3 (CN) | Disponent (DE) | Briana (RO) | Bastide (FR) | GK-Göncöl (HU) | KWS-Scirocco (DE) |
| Turkmen (TR) | Mv-Amanda (HU) | Cutter (US) | Soissons (FR) | Hallam (US) | Balada (CZ) | GK-Hattyú (HU) |
| Ravenna (IT) | Blasco (IT) | Buratino (CZ) | Libellula (IT) | Altay-2000 (TR) | Agent (US) | Dumbrava (RO) |
| Buck-Panadero (AR) | Mv-Verbunos (HU) | Klein-Flecha (AR) | | | Lupus (AT) | Cadenza (GB) |
| | Chara (AU) | | | | | |
| | Mv-Palotás (HU) | | | | | |

(Cultivars with lower values than the average distances from the group centre are listed in increasing order of their distance values).

plants in the stem elongation stage tolerate lower temperatures better than those in later phenophases [2], but the temperature maxima may be higher due to high compensatory capacity. As the results showed, the single 28˚C temperature stress at this stage of development caused less strain on the plants based on the yield related traits; there was not much deviation found between the control and the single 28˚C treatment.

The 51 wheat cultivars included in the present study were selected from 101 wheat genotypes based on their phenotypic groupings published in Balla et al. [5], where single heat stresses of various durations had been applied at three different phenological phases. In comparing the results of the two treatments across the two series of experiments, namely, a single 10-day heat stress at booting and at heading, there is a good agreement in the general tendencies, which can be summarized as follows. The heading stage was more sensitive to heat stress than the booting stage, which was primarily due to the larger decrease in the average seed number, which could no longer be compensated by the increased thousand kernel weight [36, 37]. The importance of biomass in contributing to grain yield intensified with the heat stress treatments. In both experimental series, the photosynthetic parameters, with the exception of chlorophyll content, were primarily determined by the treatment and not by the genotypic effects, and they showed only a weak correlation with the yield-related traits in various patterns across the different heat treatments. Thus, the photosynthetic parameters were not appropriate indicators for predicting heat stress tolerance, which may explain the controversial results published to date [17, 29, 38]. Finally, there was a large variation between the wheat cultivars in their response to heat stresses [31, 37, 39].

To compare the possible effects of single versus repeated heat stresses, the first stress was set at the beginning of intensive stem elongation, while the second stress was applied at one of two later phenophases, booting or full heading. For the purpose of this experiment, we chose to partition the 10-day stress period of the single stresses into 5+5 days for the repeated stresses, as the duration of stress treatment was proven to be a significant factor in the heat stress response [37, 40–42], and there were already significant differences found between a 5-and10-day heat stress [5]. This layout resulted in several changes already detectable in the average values of both the yield-related traits and physiological parameters. Averaged over the genotypes, the effects of repeated heat stress were closer to those of the control and the single heat stress at stem elongation than to the single stresses at the two later phenophases, but beyond this result, significant differences in the various magnitudes and directions could be detected across the different traits, phenophases and wheat genotypes.

Averaged over the wheat genotypes, the highest effect of repeated heat stress was detected mainly on average seed number (AS), the harvest index (HI), thousand kernel weight (TKW) and grain yield (GY), while this stress caused the smallest changes in morphological properties, such as plant height and biomass. In single stress treatments, the large reduction in AS was compensated by increased TKW, and this compensating mechanism was apparent in the repeated stresses as well. Similar positive effects of repeated heat stresses were found by Wang et al. [26]. Differences in susceptibility to early heat stress and subsequent treatments have also been well observed for photosynthetic properties [43]. It was shown that net assimilation and transpiration could be intensified in plants stressed at later developmental phases when they were also subjected to early heat stress at the stem elongation stage, which could be attributed to the greater capacities of photosynthesis maintenance under heat stress [43–45]. We found that significantly higher photosynthetic substrates (chlorophyll) were produced in response to each heat stress treatment in Exp. I. compared to the control, while in Exp. II., a reduction in chlorophyll occurred at the later heading and SE+CH stages. It seems that the higher harvest index in plants exposed to repeated heat stress had a positive relationship with the ability to retain a higher chlorophyll content, but these changes had no significant impacts on grain

yield. The findings of our study partially agree with those of Wang et al. [24], who reported that plants that received preanthesis heat stress showed much higher photosynthetic rates and Chl a/b under postanthesis high temperatures than those without preanthesis heat stress.

The reactions of plants to drought and high temperature stress are often species-specific, but there are also large differences among cultivars within a species [5, 31, 37, 39, 46]. We also found large variations among the 51 wheat cultivars included in this experiment not only in their grain yielding abilities under control conditions but also in their sensitivities to the various heat stress treatments. The genotypic reactions were proven not to be individual, and 7 distinct groups of wheat cultivars with specific response profiles could be identified at a highly significant level. When carrying out the rank correlation for the full set of 51 wheat cultivars between the phenotypic group positions in Balla et al. [5] and those of the present study, the r value was not significant (-0.247). This is not unexpected and is well explained by the different experimental setups concerning the numbers and types of heat stress treatments on which the groupings were based. However, if the members from the extreme groups of the present study are examined more closely, the overlap becomes more evident (S1 Table). Seventy-five percent of Group 2 with the lowest GY in the present study were shared with the three lowest yielding groups identified in Balla et al. [5], and only one cultivar was originally placed in one of the better groups, but not the best ones. In the cases of the three groups (Groups 5, 6 and 7) with higher grain yield in the present study, 88.3% of the cultivars originated from groups established as best and intermediate by Balla et al. [5], and only 2 cultivars (11.7%) came from the three weakest groups. To further study the genetic determinants of heat stress responses, specific crosses have been carried out between wheat cultivars showing consistent results across the two experiments, and currently, genetic populations consisting of doubled-haploid lines is under development via anther culture techniques.

One of the most important findings of this research was the identification of the large differences among the distinct groups of wheat cultivars in their responses to single versus repeated heat stresses. Almost all 7 groups were characterized by reaction patterns of different magnitudes and directions, which also depended on the phenological phases at the second cycle of heat stress. In light of the fact that the second cycle of repeated heat stress was only 5 days, as opposed to the 10-day treatment of single stresses, the question arises as to whether this phenomenon can be considered heat priming. If all the GY values of the 7 groups from the repeated heat stress treatments were significantly higher than the corresponding single heat stress treatments, then it could be stated that the difference in the duration of heat stress is the primary reason behind the results. In several cases, however, the difference was not significant, or the repeated heat stress was even proven to be more detrimental than the single stress. However, the reactions of those groups, which showed phenology stage-dependent reactions to single versus repeated heat stresses, can be more decisive in answering this question. In the case of two groups (Groups 5 and 6), the booting stage was where the significant positive effect of repeated heat stress was apparent, while there were two other groups (Groups 4 and 7) where the significant positive effect of repeated heat stress had at full heading. In all four cases, repeated stress was not significantly different from single heat stress in other stages. These specific reactions of these 4 groups were the bases for their distinct separations from the remaining groups.

These findings may emphasize the possibility of demonstrating heat priming in these experiments and, as such, may explain the controversial results published to date concerning the phenomenon of heat priming in wheat. Studies [27, 28] have failed to clearly demonstrate that early heat stress (heat priming) is responsible for improving the heat tolerance of genotypes. In some studies, in addition to heat stress, the priming effect of drought has also been studied [36]; they found that drought in the early stage could alleviate yield loss after anthesis, whereas

heat stress applied in the vegetative stage had no alleviating effect on heat stress applied in the later reproductive stage. On the other hand, there is a research underway to investigate the effects of heat priming applied to the first generation for increasing the tolerance of successive generations to post-anthesis heat stress [43]; the authors found that the trans-generation thermo-tolerance was induced by heat priming in the first generation, and it can also be an effective form of defence against high temperature stresses in the future. Our results achieved in a larger number of wheat genotypes with various geographical origins and ecological adaptation backgrounds may provide evidence that the phenomenon of heat priming exists; it is not a universal response of wheat but depends strongly on genotypes and developmental phases, in addition to the various parameters of the stress itself. Further studies are necessary to establish the conditions that may initiate heat priming and to identify their possible genetic backgrounds before practical applications in breeding can be considered realistic.

## Supporting information

**S1 Fig.** Comparison of control treatments for eighteen properties (A-B) tested in two heat stress experiments (I.-II.). A): RT—Reproductive tillers, EaL—Main ear length, MEaW—Main ear weight, MSW—Main seed weight, SEAW—Side ear weight, SSW—Side seed weight, BIOM—straw biomass, ASW—Average seed weight, DENS—Spike density (spikelet number/cm), SEN—Side ears number. B): SPIK—Spikelet number per main ear, MSN—Main seed number, SSPIK—Spikelet number per total side ears, SSN—Side seed number, GN—Grain number, GNSP—Grain number per spike, ATKW—Average thousand kernel weight, MTKW—Main thousand kernel weight; ns—not significant, *—significant at the P≤ 0.05 level.
(PDF)

**S2 Fig. Correlation matrices of various yield-related traits and photosynthetic parameters affected by different heat stress treatments in experiment I.** (*, **, *** critical r values of the correlation coefficient at 0.2732 P = 5%, 0.3541 P = 1%, 0.4433 P = 0.1%; n = 51); C—Control condition, SE—Single heat stress at stem elongation stage, B—Single heat stress at booting, and SE+B—Repeated heat stresses at stem elongation and booting stage; GY—Grain yield, HI—Harvest index, TKW—Thousand kernel weight, AS—Average seed number, PH—Plant height, LIN—Last internode length, FBIOM—Total aboveground biomass (straw + all ears), SPS—Grain number per spikelet, PN—Net assimilation, GS—Stomatal conductance, CI—Intercellular $CO_2$ concentration, E—Transpiration, CLR—Chlorophyll content.
(PDF)

**S3 Fig. Correlation matrices of various yield-related traits and photosynthetic parameters affected by different heat stress treatments in experiment II.** (*, **, *** critical r values of the correlation coefficient at 0.2732 P = 5%, 0.3541 P = 1%, 0.4433 P = 0.1%; n = 51); C—Control condition, SE—Single heat stress at stem elongation stage, CH—Single heat stress at heading, and SE+CH—Repeated heat stresses at stem elongation and heading; GY—Grain yield, HI—Harvest index, TKW—Thousand kernel weight, AS—Average seed number, PH—Plant height, LIN—Last internode length, FBIOM—Total aboveground biomass (straw + all ears), SPS—Grain number per spikelet, PN—Net assimilation, GS—Stomatal conductance, CI—Intercellular $CO_2$ concentration, E—Transpiration, CLR—Chlorophyll content.
(PDF)

**S1 Table. List of winter wheat cultivars included in the two heat stress experiments.** * in Balla et al (2019) Group 1, 2, and 3 were the best yielding, Group 4, 5, and 6 were the low yielder, while Group 7 and 8 were intermediate.
(PDF)

**S2 Table. Variance components (%) of the morphological, yield-related and physiological traits in the context of 51 wheat cultivars × treatment (single and repeated heat stress) using a general linear model in experiment I.** PH—Plant height, LIN—Last internode length, EaL—Main ear length, SPIK—Spikelet number per main ear, DENS—Spike density (spikelet number/cm), SEN—Side ears number, GN—Grain number, GNSP—Grain number per spike, SSPIK—Spikelet number per total side ears, GY—Grain yield, AS—Average seed number, ASW—Average seed weight, BIOM—straw biomass, MEaW—Main ear weight, MSN—Main seed number, MSW—Main seed weight, FBIOM—Total aboveground biomass (straw + all ears), HI—Harvest index, SPS—Grain number per spikelet, SEAW—Side ear weight, SSN—Side seed number, SSW—Side seed weight, RT—Reproductive tillers, TKW—Thousand kernel weight, ATKW—Average thousand kernel weight, MTKW—Main thousand kernel weight, EVP—Evaporation, GS—Stomatal conductance, PN—Net assimilation, CI—Intercellular $CO_2$ concentration, CLR—Chlorophyll content; ***, **, * difference significant at the 0.1%, 1% and 5% probability level.
(PDF)

**S3 Table. Variance components (%) of the morphological, yield-related and physiological traits in the context of 51 wheat cultivars × treatment (single and repeated heat stress) using a general linear model in experiment II.** PH—Plant height, LIN—Last internode length, EaL—Main ear length, SPIK—Spikelet number per main ear, DENS—Spike density (spikelet number/cm), SEN—Side ears number, GN—Grain number, GNSP—Grain number per spike, SSPIK—Spikelet number per total side ears, GY—Grain yield, AS—Average seed number, ASW—Average seed weight, BIOM—straw biomass, MEaW—Main ear weight, MSN—Main seed number, MSW—Main seed weight, FBIOM—Total aboveground biomass (straw + all ears), HI—Harvest index, SPS—Grain number per spikelet, SEAW—Side ear weight, SSN—Side seed number, SSW—Side seed weight, RT—Reproductive tillers, TKW—Thousand kernel weight, ATKW—Average thousand kernel weight, MTKW—Main thousand kernel weight, EVP—Evaporation, GS—Stomatal conductance, PN—Net assimilation, CI—Intercellular $CO_2$ concentration, CLR—Chlorophyll content; ***, **, * difference significant at the 0.1%, 1% and 5% probability level.
(PDF)

**S4 Table. Identification of the most likely cluster numbers for the 51 wheat genotypes based on the grain yield profiles achieved in the various heat stress treatments using the K-means clustering protocol.**
(PDF)

## Author Contributions

**Conceptualization:** Krisztina Balla, Ildikó Karsai, Ottó Veisz.

**Data curation:** Krisztina Balla.

**Formal analysis:** Krisztina Balla, Ildikó Karsai.

**Funding acquisition:** Krisztina Balla, Ildikó Karsai, Ottó Veisz.

**Investigation:** Krisztina Balla, Tibor Kiss, Péter Bónis.

**Methodology:** Krisztina Balla, Tibor Kiss, Péter Bónis, Tamás Árendás.

**Project administration:** Krisztina Balla, Ádám Horváth, Zita Berki, András Cseh.

**Resources:** Ottó Veisz.

**Writing – original draft:** Krisztina Balla, Ildikó Karsai.

**Writing – review & editing:** Ildikó Karsai, Ottó Veisz.

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
