## [Decision Letter · Decision Letter 0]

9 Apr 2021

PONE-D-21-06889

Single versus repeated heat stress in wheat: what are the consequences in different developmental phases?

PLOS ONE

Dear Dr. Balla,

Thank you for submitting your manuscript to PLOS ONE. After careful consideration, we feel that it has merit but does not fully meet PLOS ONE’s publication criteria as it currently stands. Therefore, we invite you to submit a revised version of the manuscript that addresses the points raised during the review process.

We look forward to receiving your revised manuscript.

Kind regards,

Aimin Zhang, Ph.D.

Academic Editor

PLOS ONE

Journal Requirements:

Reviewers' comments:

Reviewer's Responses to Questions

**Comments to the Author**

1. Is the manuscript technically sound, and do the data support the conclusions?

Reviewer #1: Yes

2. Has the statistical analysis been performed appropriately and rigorously? 

Reviewer #1: Yes

3. Have the authors made all data underlying the findings in their manuscript fully available?

Reviewer #1: Yes

4. Is the manuscript presented in an intelligible fashion and written in standard English?

Reviewer #1: No

5. Review Comments to the Author

Reviewer #1: The information provided in this paper was was of good quality and will be important to the future of this area of research. However, I found the majority of the paper hard to read due to the many grammatical errors and sentences that were structured without finesse. If this paper were to be accepted, I would suggest that the authors heavily edit and focus on restructuring sentences and some paragraphs so that they use less, but more impactful words. This will make the paper much more friendly to a wide range of readers.

6. PLOS authors have the option to publish the peer review history of their article (what does this mean?). If published, this will include your full peer review and any attached files.

Reviewer #1: No

---

## [Author Response · Author response to Decision Letter 0]

30 Apr 2021

PONE-D-21-06889: 

Manuscript title: Single versus repeated heat stress in wheat: what are the consequences in different developmental phases?

Reviewer #1: “The information provided in this paper was of good quality and will be important to the future of this area of research. However, I found the majority of the paper hard to read due to the many grammatical errors and sentences that were structured without finesse. If this paper were to be accepted, I would suggest that the authors heavily edit and focus on restructuring sentences and some paragraphs so that they use less, but more impactful words. This will make the paper much more friendly to a wide range of readers.”

Response: 

We employed the professional scientific editing service, the American Journal Experts (AJE), they re-edited and revised the complete manuscript (taking care to correct grammatical errors and sentences and comply with formal requirements) according to the submission guideline of PLOS ONE. For incomprehensible sentences, we also focused on re-editing them. Below you find the Editing Certificate of the manuscript PONE-D-21-06889.

---

## [Decision Letter · Decision Letter 1]

10 May 2021

Single versus repeated heat stress in wheat: what are the consequences in different developmental phases?

PONE-D-21-06889R1

Dear Dr. Balla,

We’re pleased to inform you that your manuscript has been judged scientifically suitable for publication and will be formally accepted for publication once it meets all outstanding technical requirements.

Kind regards,

Aimin Zhang, Ph.D.

Academic Editor

PLOS ONE

Additional Editor Comments (optional):

Reviewers' comments:

Reviewer's Responses to Questions

**Comments to the Author**

1. If the authors have adequately addressed your comments raised in a previous round of review and you feel that this manuscript is now acceptable for publication, you may indicate that here to bypass the “Comments to the Author” section, enter your conflict of interest statement in the “Confidential to Editor” section, and submit your "Accept" recommendation.

Reviewer #1: All comments have been addressed

2. Is the manuscript technically sound, and do the data support the conclusions?

Reviewer #1: Yes

3. Has the statistical analysis been performed appropriately and rigorously? 

Reviewer #1: Yes

4. Have the authors made all data underlying the findings in their manuscript fully available?

Reviewer #1: Yes

5. Is the manuscript presented in an intelligible fashion and written in standard English?

Reviewer #1: Yes

6. Review Comments to the Author

Reviewer #1: I found the manuscript to have been adequately revised and I very much enjoyed reading it. The editing that has been carried out to improve sentence structure and grammar was well worth the effort and I appreciate what the authors have accomplished. The manuscript gives valuable insight into the importance of research into heat priming of wheat and improves the understanding of some mixed messages that have arisen from this type of research over time.

7. PLOS authors have the option to publish the peer review history of their article (what does this mean?). If published, this will include your full peer review and any attached files.

Reviewer #1: **Yes: **Rebecca J Thistlethwaite

---

## [Editor Report · Acceptance letter]

14 May 2021

PONE-D-21-06889R1 

Single versus repeated heat stress in wheat: what are the consequences in different developmental phases? 

Dear Dr. Balla:

I'm pleased to inform you that your manuscript has been deemed suitable for publication in PLOS ONE. Congratulations! Your manuscript is now with our production department. 

Kind regards, 

on behalf of

Prof. Aimin Zhang 

Academic Editor

PLOS ONE